# Two Faces of Autophagy in the Struggle against Cancer

**DOI:** 10.3390/ijms22062981

**Published:** 2021-03-15

**Authors:** Anna Chmurska, Karolina Matczak, Agnieszka Marczak

**Affiliations:** 1Doctoral School of Exact and Natural Sciences, University of Lodz, Banacha Street 12/16, 90-237 Lodz, Poland; 2Department of Medical Biophysics, Faculty of Biology and Environmental Protection, Institute of Biophysics, University of Lodz, Pomorska Street 141/143, 90-236 Lodz, Poland; karolina.matczak@biol.uni.lodz.pl (K.M.); agnieszka.marczak@biol.uni.lodz.pl (A.M.)

**Keywords:** anticancer therapies, cancer stem cells, cell death, autophagy modulation, apoptosis

## Abstract

Autophagy can play a double role in cancerogenesis: it can either inhibit further development of the disease or protect cells, causing stimulation of tumour growth. This phenomenon is called “autophagy paradox”, and is characterised by the features that the autophagy process provides the necessary substrates for biosynthesis to meet the cell’s energy needs, and that the over-programmed activity of this process can lead to cell death through apoptosis. The fight against cancer is a difficult process due to high levels of resistance to chemotherapy and radiotherapy. More and more research is indicating that autophagy may play a very important role in the development of resistance by protecting cancer cells, which is why autophagy in cancer therapy can act as a “double-edged sword”. This paper attempts to analyse the influence of autophagy and cancer stem cells on tumour development, and to compare new therapeutic strategies based on the modulation of these processes.

## 1. Introduction

Autophagy is a process whose primary function is to maintain cellular metabolism and survival during starvation and stress, thus preventing cell death. Although apoptosis and autophagy are aimed at different targets and have distinct morphological and biochemical characteristics, there are several interactions between them [1]. The signalling pathways responsible for the interactions between autophagy and apoptosis have become the subject of intense research in recent years. Importantly, particular attention is being paid to how autophagy should be approached and whether it contributes to cancer treatment. The importance of autophagy in cancer therapy is particularly relevant. Autophagy is generally considered to be a cancer-preventing mechanism that occurs in two ways: sovereign and dependent [2]. It can maintain normal cellular homeostasis by removing oncogenic proteins or damaged organelles from cells. This contributes to the protection of cells against the development of neoplastic lesions. Furthermore, autophagy works closely with the immune system to provide a noncellular form of defence against cancer. Unfortunately, enhanced autophagy facilitates the survival and growth of malignant neoplastic lesions. Thus, the role of autophagy in tumour formation is multifaceted: it promotes tumour cell survival by providing essential components for growth, but it is also involved in the regulation of tumour cell migration and invasion [3]. This review focuses on how autophagy may contribute to cancer development, how it affects CSCs and which autophagy-related treatments currently exist.

## 2. Apoptosis

In 1964, Lockshin and Williams published their research “Programmed cell death—I. Cytology of degeneration in the intersegmental muscles of the Pernyi silkmoth”. In this publication, the death of specific cells during silkworm metamorphosis was observed. This type of cell death was called “programmed cell death”, because cells were dying according to the insect’s “instruction” [4]. The term “apoptosis” was used for the first time by Kerr, Curie and Wyllie in 1972 [5,6,7], although Flemming discovered the process and called it “chromatolis” in 1885 [8]. Programmed cell death is commonly called “apoptosis”, a term derived from ancient Greek, meaning “falling down” [7]. This process is genetically regulated starting from the embryonic stage to whole ontogenesis. Apoptosis plays a primary role in the remodelling of the developing tissues, as well as in the proper functionality of an immune system [9,10]. 

Due to progress in the field of electron microscopy, it became possible to determine the various morphological changes which occur during apoptosis. The most unique features of apoptosis are pyknosis and cell shrinkage. During the shrinkage process, cell size is reduced, while cytoplasm and cell organelles undergo condensation. Cariopyknosis, which is typical for apoptotic cells, is the consequence of chromatin condensation [11]. The hematoxylin-eosin colouring revealed that apoptotic cells resemble an oval mass with thick, dark cytoplasm and dense, purple, fragmented nuclear chromatin. At the stage of condensation, chromatin aggregates under the nuclear membrane [12]. With cytoplasm condensation progression and reduction of cell size, cell membranes form bubble-like structures. At the same time, two other processes are taking place, namely, karyorrhexis and budding, during which disassociation of cell fragments occurs. Furthermore, it should be mentioned that budding is the process in which cell fragments disassociate [13]. Apoptotic bodies created in this process contain cytoplasm with tightly packed cell organelles and with fragments of a nucleus. Subsequently, apoptotic bodies undergo phagocytosis, a process in which macrophages, parenchymal cells and histiocides take part (Figure 1).

The most important feature of apoptosis is the absence of an inflammatory reaction in the proximity of the dying cells, caused by the facts that: During apoptosis, cells do not disintegrate;There is no secondary necrosis, due to the immense pace of the phagocytosis;Phagocytes (such as macrophages, histiocytes, interstitial cells) do not release anti-inflammatory cytokines [14].

### 2.1. Mechanism of Apoptosis

The mechanism of programmed death is complex, involving a cascade of molecular factors depending on ATP. During apoptosis, the following phases can be distinguished:Induction phase,Effector phase,Degradation phase [13].

One can distinguish two main pathways of apoptosis:Extrinsic,Intrinsic.

The activation of the hydrolytic enzymes, especially protease and nuclease, are the direct cause for the disintegration of cell structures during apoptosis. Beside activation of hydrolytic enzymes, inactivation of cell repairing systems occurs. The activation of endonuclease and DNA fragmentation are the initial signals of apoptotic death. The synthesis of specific cysteine proteases (caspases) in a cell is typical of programmed cell death (PCD). The high-level phosphatidylserine in the outer layer of the plasma membrane is another important feature of apoptosis [15].

The caspases, cysteine proteases, are activated independently from the apoptotic pathway. Taking into consideration the stage of apoptosis, during which they are activated, two types of caspases can be distinguished: initiators (−2, −8, −9, −10, −12) and executioners (−3, −6, −7) [16]. These enzymes are formed in a cell and stored there in an inactive form. Caspases are activated when apoptosis is initiated. They carry a signal which activates transcription factors such as NF-κB and AP-1. Consequently, proteins with abnormal structure are formed; this may lead to cell death [17,18].

#### 2.1.1. Extrinsic Pathway

The extrinsic pathway of apoptosis is induced by a limited number of growth factors or nutrients, as well as a local increase in hormone and cytokine concentrations. Furthermore, the factors which activate extrinsic pathway are chemical compounds including physical factors and cytostatics. The proapoptotic signal is transmitted via TNF family receptors. After binding of the appropriate ligand to the receptor for example FasL/TNF-α ligand to FasR/ TNFR1 one induces transmission of signal through FADD’s death domain to caspase-8. Subsequently, the activation of the death-inducing signalling complex (DISC) results in inducing caspase-8 and sending a signal to the effector caspases (−3 and −7) [19,20].

#### 2.1.2. Intrinsic Pathway

The intrinsic pathway of apoptosis involves mitochondria. As a result of an increased concentration of calcium ions (Ca^2+^), ROS, hypoxia, hormone and growth factors deficiency, the apoptosis process is activated. The aforementioned factors influence the activation of caspases, the permeability of mitochondrial membranes and interactions with proteins mediating cell decay. This leads to release of cytochrome c from the mitochondrial space and activation of the caspases. The relevant structural element related to cytochrome c release from the mitochondria is called mitochondrial permeability transition pore (PTP). Cytochrome c binds to Apaf-1 protein factor and procaspase-9. Consequently, a three-component system called apoptosome is created. The formation of apoptosome requires energy from ATP hydrolysis. Beside cytochrome c, over 40 proteins are involved in the activation of the intrinsic apoptotic pathway, e.g., apoptosis inducing factor (AIF). This is transferred to the nucleus during initiation phase. AIF and endonuclease G are released from mitochondria and translocated to nucleus. These proteins affect nucleus apoptotic changes [21,22,23].

Both of apoptosis signalling pathways are linked, and can influence each other via molecules [24]. Interestingly, extrinsic and intrinsic pathways come together in the same phase, i.e., the effector phase [25,26].

## 3. Autophagy 

The word autophagy originates from Greek language: auto—self, and phagy—eat. The term “autophagy” was used for the first time by French scientist Anselmier in 1859. In his article, Anselmier described the effect of fasting on mice. A century later, Christian De Duve described this process at Ciba Foundation Symposium on Lysosomes in 1963 [27,28]. Huge interest in autophagy occurred in 2016, when a Nobel Prize was awarded to Yoshinori Ohsumi for discovering ATG genes, i.e., genes control and regulate autophagy in yeast [29].

### 3.1. Autophagy Mechanism

Autophagy is defined as a conservative catabolic process which is observed in all eukaryotic organisms. This process, as a cytoprotective mechanism, is triggered by various incentives, such as nutrient deficiencies, oxidative stress, or hypoxia [30,31]. The primary role of this process is to maintain the cell’s metabolism during starvation, and to protect it against the accumulation of damaged proteins or toxins when faced with stress factors [32]. Autophagy can be divided into three types on the basis of how cell elements are delivered to lysosomes and vacuole: (1) The process of microautophagy eliminates compounds dispersed in the cytoplasm, small organelles and fragments of cellular nucleus. During this process, the substrate is surrounded by a lysosome membrane and transported to its interior by endocytosis [33]; (2) Chaperone-mediated autophagy (CMA) is a mechanism involving chaperons, which task is to capture e.g., cytosolic proteins of inappropriate conformation. A characteristic feature of CMA is that each substrate protein contains a KFERQ theme in its amino acid sequence. This sequence is responsible for directing the protein to the lysosome and is recognised by the heat shock proteins [34]; and (3) Macroautophagy, commonly called autophagy, during which C-shaped double-membrane surrounding cytoplasm marks the beginning of autophagy. Both of membrane ends get longer, enclosing various elements of a cell, including cell organelles such as damaged mitochondria, ribosome and endoplasmic reticulum. The resulting vesicle is known as an “autophagosome” [35]. After fusion with lysosome, autophagolysosome degradation of cargo is initiated. The products of degradation, for example, amino acids, carbohydrates and fatty acids, are basic components of macromolecular substrates. These products are transported to the cytoplasm in order to meet cellular metabolic needs, or are used to repair damage [36,37]. Although the forms of autophagy differ, substrates are delivered to lysosomes in all types. Of the three types of autophagy, macroatophagy is the best-known form [38]. During autophagy, the following phases are distinguished (Figure 2):Autophagy initiation,Phagophore nucleation,Elongation,Fusion with lysosome,Degradation [33].

#### 3.1.1. Autophagy Initiation

This process starts with the activation of ULK1 complex. The complex includes ULK1, ATG13, ATG101 and FIP200/RB1CC1 protein [39]. The main regulator of the complex activity and at the same time whole autophagy process is mTOR kinase. The mammalian target of rapamycin (mTOR) is a sensor of nutrient availability in the cell environment. With nutrient high supply, mTOR inhibits autophagy by phosphorylation of ULK1/2 kinases, thus blocking the activity of entire ULK/Atg13/Atg101/FIP200 initiation complex. Under starving conditions, mTOR dissociates from ULK1/2, causing them to dephosphorylate, resulting in cell growth promotion by protein catabolism inhibition. While mTOR is inactivated, autophagy is activated by the formation of ULK complex. However, mTOR kinase is not the only regulator of ULK complex activity. When there is an energy deficit in a cell, it can be activated by AMPK kinase directly and indirectly by inhibiting mTOR kinase activity.

#### 3.1.2. Nucleation

A complex, which includes the class III PI3K kinase, plays an important role at the nucleation stage, i.e., in the early stage of phagophore formation. The complex includes VPS34, VPS15, Beclin-1, ATG14. Its activity of the complex is important in providing phospholipid, phosphatidylinositol triphosphate (PI3P), in place of forming isolating membranes. The interaction with the UV irradiation resistance-associated gene (UVRAG) protein is necessary to create this complex. The complex produces the phosphatidylinositol-(3,4,5)-triphosphate necessary for the elongation of phagophosphorus [40,41,42].

#### 3.1.3. Elongation

ATG12-ATG5-ATG16L and ATG8 systems are necessary for the correct course of the phagophore elongation stage [43]. The first complex requires the presence of four proteins: ATG5, 7, 10, 12. The ATG12 and ATG7 proteins form an unstable connection, then ATG12 is transferred to ATG10, which is the equivalent of the enzyme E2. Subsequently, ATG12 protein is attached to ATG5. In the next stage the ATG12-ATG5 complex is bound to the ATG16 protein. The second ubiquitous similar complex, necessary for the formation of autophagosome is LC3-II-PE. LC3, a light chain of microtubule-bound proteins, is the suckling orthologist of the yeast protein ATG8. ATG3, 4, 7 proteins and phosphatidylethanolamine participate in the creation of second ATG8/LC3 conjugant system. The LC3 protein, found in the cytosol, is synthesised as a pro-LC3 precursor which, as a result of a proteolytic cut by ATG4, is converted to the LC3-I form. The action of ATG7 and ATG3, acting as ubiquitin-like enzymes and the reversible bound of LC3-I with the amine group phosphatidylethanolamine (PE), produces a mature form LC3-II [44,45].

#### 3.1.4. Fusion with Lysosome and Degradation

When the elongating ends of the phagophore create a bubble, called “autophagosome”, the next stage of autophagy, i.e., the binding of the autophagosome and lysosome, begins. The result is a bubble surrounded by a single membrane, in which final destruction of its content takes place, under the influence of lysosomal enzymes. The presence of LAMP-2 protein, RAB-GTPases and SNAREs is necessary for the proper course of the maturation process [43].

## 4. Two Faces of Autophagy

Autophagy plays a dual role in the development of cancer; it can inhibit further disease development or protect cancer cells, causing tumour growth. This phenomenon is called “autophagy paradox” [46]. On the one hand, autophagy degrades some cellular elements, providing substrate for biosynthesis. On the other hand, excessive activity of this process can lead to exorbitant degradation and ultimately to cell death by apoptosis. Besides the phenomenon itself, autophagy can promote or inhibit drug resistance [47,48].

More and more researchers have reported that autophagy may play significant role in chemoresistance in various types of cancer. Depending on the type of cancer and treatment method, increased autophagy activity may promote and inhibit drug resistance [49,50].

Some types of cancer use autophagy to adapt to hypoxia or nutrient deficiency. During anticancer therapies, induction of autophagy as a protective and pro-life was observed in cancer cells. Furthermore, hypoxia can induce autophagy, which suggests that this process is involved in acquisition of resistance [51]. 

Autophagy promotes cell survival and the development of drug resistance. Increased autophagy activity and decreased apoptosis activity have been observed in multiple myeloma resistant to DOX. It has been shown that cytoprotective autophagy was induced in MDA-MB-231 breast cancer cells treated with DOX. This research indicated that autophagy could protect cancer cells, which may result in lower sensitivity to treatment. Therefore, autophagy in anticancer therapy acts as a “double-edged sword”. The attempts to inhibit or activate autophagy, based on its inconsistent character, can be an effective anticancer strategy [52].

Only a few human studies have shown the impact of autophagy on the development of cancer. The mutation of the *BECN1* and *ATG7* genes observed in human ALDH+ breast cancers resulted in reduced proliferation of cancer stem cells (CSCs) and loss of their ability to self-renew [53].

The conditions of increasing metabolic stress, which occur in intensively dividing cancer cells as a result of insufficient supply of oxygen and nutrients, together with an extremely high demand for energy substrates, require the activity of autophagy as a mechanism for keeping it alive. However, it turns out that autophagy in carcinogenesis can both enable cancer cells to survive, as well as act in the opposite direction, contributing to their elimination.

One of the main objectives of research conducted to develop effective anticancer therapies is to increase the susceptibility of cancer cells to apoptosis activation. However, it turns out that cancer resistance to treatment is related not only to malfunctioning of programmed cell death, but also to the protection under stress condition provided by the process of autophagy. Apparently, the link between apoptosis and autophagy affects the fate of the cell. While the activation of autophagy is (at least initially) associated with the inhibition of apoptosis (allowing the cell to survive), vice versa: proapoptotic signalling inhibits autophagy [54,55]. Both in normal and neoplastically transformed cells, these processes are connected by common signal pathways and so-called molecular switches—proteins that can directly regulate both processes [56]. 

Autophagy was considered a suppressor of carcinogenesis when the *BECN1* gene was tested. Deletion of *BECN1* gene occurs in 40% cases of prostate cancer, 50% cases of breast cancer and 75% cases of ovarian cancer. Moreover, decreased expression of *BECN1* was observed in other kinds of cancer, including colon, brain and cervical ones. The data suggest, that abnormal course of autophagy is related to cancer development [57,58,59]. The ectopic expression of this gene reduces both in vitro cancer cell proliferation and cancer potential in vivo. The studies indicated that mice with monoallelic deletion of the gene encoding becline-1 (Becn1+/−) showed a significant increase in the incidence of spontaneous tumours in compared to mice having both wild alleles [60].

The relationship between impaired autophagy and tumour development is best evidenced by inhibition of the latter process in mice (the cancer development was caused by loss of *BECN1* gene). In MCF-7 cell line, showing low expression of Bcl-1, proliferation has been inhibited after transfection of *BECN1*. According to the researches, the reduced expression of BECN1 does not prognose well for sick patients [58]. 

The essential factor connecting autophagy and apoptosis is beclin-1. This protein is associated in the complex with antiapoptotic proteins belonging to the BCL-2 family. Beclin-1 binds to these proteins through its domain BH3, preventing it from participating in the initiation of autophagy [61,62]. Autophagy allows the cell to restore homeostasis and avoid cell death. An important issue in the interaction between autophagy and apoptosis is that regardless of the location in the cell, the binding of beclin-1 does not affect the antiapoptotic effect of Bcl-2 proteins. On the other hand, the location in the cell affects the ability of Bcl-2 proteins to bind beclin-1. The nutrient deficiency autophagy factor 1 may affect this ability. NAF-1 probably stabilises the Bcl-2-beclin-1 complex in the endoplasmic retina. In case of conditions stimulating autophagy, such proteins as BAD, BNIP3, NIX, NOXA and PUMA bind the antiapoptotic Bcl2 protein, which causes the release of beclin-1, which enables the formation of the PI3K complex [53,63].

The existence of beclin-1 complexes with Bcl-2 proteins is regulated on several levels: by the JNK kinase in response to the availability of nutrients, by the DAPK kinase, HMGB protein activated under oxidative stress conditions in cell or proapoptotic proteins Bad and Bax [64,65,66].

The previously mentioned protein, HMGB1, protects beclin-1 and ATG5 from being cut by calpains. As a result, it prevents the formation of proapoptotic fragments of these proteins. HMGB1 protein, prevents the induction of apoptosis while maintaining the process of autophagy, thus contributing to the reduction of tissue damage. HMGB1 protein can therefore be considered a “molecular switch” between autophagy and apoptosis [67].

As a result of cutting by the calpains, the ATG5 protein shows proapoptotic properties. In mitochondria, the N-terminal fragment of ATG5 binds to the Bcl-XL protein and thus participates in mitochondrial-dependent apoptosis induction. When ATG5 is not transferred to the mitochondrion, autophagy is initiated in the cells [68]. Interestingly, theATG5 protein is necessary for the initiation of p53 induced apoptosis. Moreover, ATG5 also interacts with the FADD protein, but this interaction does not affect the formation of autophagosome, it may be associated with apoptosis inhibition independently of autophagy. As one of the main proteins of the autophagic pathway, ATG5 may be involved in positive and negative regulation of apoptosis [69].

The caspases participating in the process of apoptosis take part in the degradation of proteins important for autophagy, such as Beclin-1, AMBRA1, ATG3, ATG4, ATG5, ATG7, and p62. Furthermore, some proautophagic proteins after cleaving by caspases may transform to pro-apoptotic proteins and induce apoptosis [70,71].

The FLIP protein, which blocks the initiation of apoptosis at the level of death receptors, also controls the process of autophagy. Through DED domains, FLIP protein recognises and connects to the protein Atg3, and as a consequence prevents the conjugation of Atg3 protein with LC3 protein and blocks autophagy at the stage of creating the autophagosome [68].

Changes in other genes have been shown to affect autophagy. Mutation in *UVRAG* gene has been found in gastric and colon cancer cells. *UVRAG* gene encodes a protein which interacts with Bcl-1. It was observed that when this gene was overexpressed in colon cancer cells, proliferation was reduced and tumour growth was much slower [72].

It has been found that genetic modifications of *ATG* genes, which are observed in human cancer cells, can lead to the development of cancer in mice. Frameshift-type mutations in *ATG12*, *ATG2B*, *ATG9B* and *ATG5* occur in 25% cases of gastric and colon cancers. Moreover, the research showed that *ATG5* mutations lead to autophagy inhibition by interrupting *ATG5-ATG16L1* interactions [73].

Protein genes with altered levels of expression are considered as potential cancer markers. It has been shown that high levels of expression of several *ATG* genes are associated with high survival of patients, but unfortunately it also turned out that some ATG proteins are considered as unfavourable markers of prognosis [74]. Depending on the type of cancer there are different levels of episodes of *ATG* genes, markers of high risk of recurrence of colorectal are: *ATG16L2*, *CAPN2* and *TP63* upregulation, ATG5 downregulation and other genes associated with autophagy, i.e., *SIRT1*, *RPS6KB1*, *PEX3*, *UVRAG* and *NAF1*. Other factors that contribute to the development of gastric cancer are *ULK1*, *BECN1*, *ATG3* and *ATG10* [75,76].

There is evidence suggesting that expression of the genes related to autophagy is limited in cancer cells. This is a huge problem, because autophagy plays a crucial role as a cancer suppressor during oncogenesis. Induction of process can help to fight the disease [77,78].

To sum up, there is a network of relationships between autophagy and apoptosis, and the signals circulating within it will determine the fate of the cell. Therefore, it should be remembered that every dysfunction of one of these processes will have its expression in the dysfunction of the other.

## 5. The Influence of Autophagy on Anticancer Therapy

### 5.1. Autophagy Inducers 

Moreover, apoptosis may increase resistance to applied therapies. On this account, new strategies to improve anticancer therapies have been sought. The induction of cell death via autophagy by anticancer drugs or autophagy inducers may be an attractive therapeutic strategy for the elimination of cancer cells [49]. 

Cytostatic drugs and radiation were shown to cause autophagy. For example, autophagy may be induced by imatinib, tyrosine kinase inhibitor BCR ABL, or cetuximab, monoclonal antibody directed against epidermal growth factor receptor (EGFR) and proteasome inhibitors. Several other compound including seocalcitol induce autophagy in cells [79,80]. Additionally, pharmacologically activated autophagy by analogues rapamycin and metformin could inhibit cancer transformation [81]. 

It is worth noting that metformin, which is a synthetic guanidine derivative, is used in the treatment of diabetes. This compound has been recognised as one of the key drugs in the treatment of patients with type 2 diabetes [82]. Metformin can induce a reduction in cell proliferation through the activation of AMPK, which leads to the inhibition of mTOR and consequently to the activation of apoptosis or cell cycle arrest. Nevertheless, AMPK activation also promotes the induction of autophagy through direct phosphorylation of ULK1 and parallel inhibition of the mTORC1 complex through phosphorylation of TSC2 and Raptor [83,84]. Furthermore, metformin induces autophagy by accumulating LC3-II and reducing p62 protein levels, thereby promoting TRAIL-induced apoptosis in TRAIL-resistant lung cancer cells [85].

Rapamycin, which is a natural mTOR inhibitor, as well as its analogues, have found use as potential drugs in anticancer therapy. There is now a growing number of reports suggesting that rapamycin analogue mTOR inhibition is linked to autophagy activation. The findings indicate that rapamycin and its analogues, at certain concentrations, show low toxicity, with the ability to inhibit the development of cancer cells [86]. An interesting point is that, according to studies, rapamycin can potentiate the anticancer properties that doxorubicin has and, when combined, can inhibit cancer cell growth and lead to cell death by impacting mTOR/p70S6K signalling [87].

Prevention of disease development by activating autophagy is a very interesting research topic. However, it is not yet clear whether improving autophagy is a potential chemopreventive and/or chemotherapeutic operation principle of these agents [58].

### 5.2. Autophagy Inhibitors

Despite many advances in the fight against cancer, many types of cancers do not respond satisfactorily to existing therapies. The latest research shows that inhibition of autophagy creates opportunities to strengthen the action of anticancer drugs. Autophagy may stop apoptosis caused by DNA damage inducing agents or hormonal therapies. Studies have indicated that autophagy can be inhibited by siRNA, which silences the genes involved in this process (*ATG5*, *ATG6/BECN1*, *ATG10* and *ATG12*) or by inhibitors (3-methyladenine, hydroxychloroquine, bafilomycin A1) [88].

Inhibitors of autophagy can act early or late in this process, with the help of regulators including ULK1, VPS34 and ATG4B (Table 1). The group of inhibitors that act in the early phase of autophagy includes 3-methyloadenine and wortmannin. The aim of these compounds are class III PI3K kinase [89]. Comparing these two compounds, wortmannin has a better effect than 3-methyloadenine, it binds irreversibly class III PI3K kinase. Other PI3K inhibitors are LY294002 and SF1126 (analog of LY294002) [90,91]. 

The serine/threonine kinases ULK1 and ULK2 constitute possible targets for anticancer therapy. It has been shown that inhibition of ULK1 induced apoptosis and affected the size of the tumour. However, inhibitors ULK1 influence on ULK2. The most closely studied compound is SBI-0206965, which, although it has strong ULK1 inhibitory properties, also has off-target properties [92,93]. It inhibits autophagy and most importantly induce apoptosis in NSCLC [94], RCC [95] and neuroblastoma cells [96]. Curiously enough, structural studies have recently shown that ULK1 inhibitors that exhibit inhibitory properties can also inhibit Aurora kinase and ULK2. Beside SBI-0206965, there are other candidates for autophagy inhibitors such as MRT68921 and MRT67307 [97] (Table 1).

As mentioned earlier, an interesting target for autophagy inhibition is VPS34; such inhibitors include VPS34-IN1 [98], SAR405 [99], compound 13 [100], SB02024 [101]. SB02024 as a potential autophagy inhibitor shows effective inhibitory properties, in addition it also has a very good pharmacokinetic profile and shows synergistic effects with other therapies, making it suitable for subsequent characterisation as a potential candidate for clinical therapy [101].

At the late stage of autophagy, chloroquine, hydroxychloroquine and bafilomycin A1 inhibitors are used. Bafilomycin A1 is a specific vacuolar-ATPase inhibitor which inhibits the lysosomal transport of protons and consequently inhibits autophagic flux [57]. 

Chloroquine (CQ) and hydroxychloroquine (HCQ) are lysosomotropic compounds, preventing acidification of lysosomes. These drugs are used to treat malaria and rheumatoid arthritis, due to their ability to overcome blood-brain barrier. It has been shown, CQ and HCQ can increase cytotoxic effect of chemotherapy [102,103]. Furthermore, it was proven the combination of therapy with CQ and HCQ strongly induces cell death. Recently, it has been shown Lys05, which is analog of chloroquine, is more influential than hydroxychloroquine [104]. Apart from these compounds, there are other lysosomal inhibitors such as DQ661 [105], VATG-027, VATG-032 [106] and melfquine [107].

**Table 1 ijms-22-02981-t001:** Classification of autophagic inhibitors according to their location of action.

Target	Drug	References
PI3K	3-methyloadenine	[108]
	LY294002	[90]
	SF1126	[91]
Lysosome	Chloroquine	[109]
	Hydroxychloroquine	[109]
	Lys05	[104]
	DQ661	[105]
	VATG-027	[106]
	VATG-032	[106]
	melfquine	[107]
ULK	MRT67307	[97]
	MRT68921	[97]
	SBI-0206965	[92,93]
Vacuolar-ATPase inhibitor	Bafilomycin A1	[109]
Vps34	SAR405	[99]
	VPS34-IN1	[98]
	compound 13	[100]
	SB02024	[101]

## 6. Autophagy in Cancer Treatment

Autophagy modulation and combination treatment provide prospects in the fight against cancer. The combinations of autophagy inhibitors/activators with chemotherapy or radiotherapy have shown better results in cancer treatment. To date, the only drugs currently used in patients in a treatment-directed manner to target autophagy are CQ and its derivative hydroxychloroquine (HCQ) [101]. Victor Bedoya first identified CQ as a potential anticancer agent by finding CQ toxicity against lymphoma and melanoma cells in 1970 [110]. In contrast, Murakami et al. were the first to unequivocally identify the ability of CQ to inhibit autophagy [111]. Promising results have come from early clinical trials using CQ/HCQ in combination with other treatments, demonstrating that the treatment goals of inhibiting autophagy can be achieved with minimal toxicity. For example, the use of trastuzumab and chloroquine have almost completely inhibited in HER2-positive breast cancer that was previously resistant to trastuzumab [112]. Moreover, autophagy inhibition in 45-66% of patients occurred during combination therapy HCQ with radiotherapy and chemotherapy, which it was at the same time [113]. 

In addition, antitumour effects have been observed, e.g., patients with glioma who received CQ in combination with radiotherapy and temozolomide (TZD) showed a threefold increase in median survival compared to patients in the control group [114]. While 41% of melanoma patients treated with CQ and TZD showed a partial response or disease stabilisation, 84% of patients treated with CQ and radiotherapy with brain metastases had a one-year survival compared to 55% of patients treated with radiotherapy alone [115,116].

The issue of autophagy dependence is important because in autophagy-dependent tumour cells, there may be synergism between autophagy inhibitors and other drugs. In contrast, in autophagy-independent cancer, they may act antagonistically [117]. Especially autophagy-dependent are brain tumours with BRAFV600E mutations [118]. Vemurafenib and CQ combination therapies reduce tumour cell viability. These findings have prompted the introduction of the use of CQ in patients with vemurafenib-resistant brain cancer [119]. Although initially promising, these studies found that autophagy inhibition was sufficient to kill BrafV600E-positive CNS cancer cells, but this was not the case for counterparts expressing wild-type BRAF [118].

Unfortunately, in selected cases inhibition of autophagy is contraindicated. A recent study showed that inhibition of autophagy induced the resumption of abnormal expression of 6-phosphofructo-2-kinase/fructose-2,6-biphosphatase 3 (PFKFB3) in mouse breast cancer stem cells (BCSCs), which in turn contributed to their reactivation and proliferation and subsequent metastasis [120]. The above results emphasise the requirement for patients and drugs to be properly selected in order to obtain the greatest benefit from autophagy inhibition and to limit potential side effects. Presumably due to a number of factors, including the nearly complete absence of biomarkers to identify patients suitable for treatment, the fact that there are no specific and more potent autophagy inhibitors, and incompletely understood potential resistance mechanisms. The clinical drugs currently in use, including CQ and HCQ, are unfortunately not specific for autophagy [121]. 

The development of nanotechnology may not only allow the delivery of anticancer agents directly to the tumour, but may also allow for the encapsulation of autophagy inhibitors, which may result in more efficient treatment. For instance, treatment with CQ may allow much better accumulation of drugs in cancer cells [122,123]. Taking advantage of advances in nanotechnology, it was possible to encapsulate miR-375, an autophagy inhibitor, and sorafenib in lipid-coated calcium carbonate nanoparticles [124]. Using this approach, there was significant inhibition of autophagy and increased antitumour activity of sorafenib in liver cancer. Another group of researchers developed Dox/wortmannin-regulated micelles exerting remarkable anticancer effects in melanoma and breast cancer by inhibiting autophagy [125].

Given that autophagy inhibitors can be used in combination with currently commonly used chemotherapeutics or other available anticancer therapies, they should be considered an extremely promising form of cancer control. For future clinical applications, further work is needed to identify the significance of autophagy induced by cancer therapy, to design a more efficient drug delivery system and to discover potential new autophagy inhibitors.

## 7. Autophagy in Stem Cells

Stem cells (SC) are a unique class of cells in the living organism. These are immortal, nonspecialised cells which, in the appropriate microenvironment, have unique plasticity and the potential to differentiate into any kind of cell [126]. The most characteristic feature of stem cells is their ability to self-renew (multiple symmetrical or asymmetrical divisions without differentiation and aging) and multi-directional differentiation into organ-oriented specialised progeny cells. Especially important is the ability of stem cells to asymmetrical divisions, during which one of the progeny cells remains a stem cell and the other begins the process of differentiation [127,128]. However, this definition of stem cells is rather simplified due to the fact that there are many types of stem cells that differ in either their proliferative potential or their ability to differentiate. It shows that as long as the stem cells are heterogeneous, they are difficult to put together in a single, precise definition [129].

## 8. The Unique Type of Stem Cells—Cancer Stem Cells (CSC)

In recent years, it has been unequivocally demonstrated that not all cells are functionally equivalent in tumours. Within each individual tumour, there is significant genetic epiheterogeneity [130,131]. Cancer stem cells are a population of undifferentiated cells which can be separated by the expression of the appropriate surface markers [129]. These cells are characterised by increased oncogenesis capacity and their ability to induce tumour formation, and may also be responsible for tumour metastasis [132]. Cancer stem cells have a wide range of common features with normal stem cells. They demonstrate the ability to self-renew, proliferate, differentiate and multiply (Figure 3) [133].

An important factor which directly affects the maintenance of a constant number of stem cells is their microenvironment, termed “niche” [134]. The niche is responsible, among other things, for whether the cell will be self-renewing or differentiated into another cell type. The niche structure is complex. It is composed of the mesenchymal cells of the immune system, a network of blood vessels and extracellular matrix components [135]. In order to sustain their properties and ability to regenerate, stem cells require signals from the niche cells in which they occur. The niche also constitutes a protection against the uncontrolled spread of stem cells within the organism [136]. Cancer stem cells are probably controlled by the same regulation within the niche [137]. The microenvironment of cancer stem cells is usually located in the region of blood vessels, which facilitates their metastasis. It has been suggested that the deposition of cells that are metastasised to other organs may be the beginning of the formation of a new microenvironment in the area of the metastasis that starts to initiate the formation and growth of secondary tumours in other tissues (Figure 4) [138].

Stem cells have very limited or no possibility of self-renewal, which suggests that the population of the cancer stem cells may be responsible for the development and propagation of the cancer. 

## 9. The Role of Autophagy in the Maintenance of Stem Cell Property

In an effort to demonstrate their characteristic and unique properties (multipotentiality, differentiation and self-regeneration), all stem cells must strictly control the intracellular level of each individual protein and the intensity of ATP production. Such control of cell metabolism is considered to be the basic mechanism for maintaining stem cells in the state of quiescence [139]. Maintaining the balance between the processes of stem cell differentiation and their quenching is crucial for keeping the stem cells in a proper state. Increased differentiation process may deplete the pool of stem cells, leading to their aging. Furthermore, an increased differentiation of stem cells may lead to the development of cancer. Thus, maintenance of proper quality control mechanisms is necessary to sustain homeostasis and the ability to respond appropriately to damage, environmental stress or differentiation signals, allowing for proper tissue regeneration.

## 10. Autophagy in Cancer Stem Cells

Autophagy is one of the most crucial processes in maintaining the activity and aggressiveness of cancer stem cells. The preservation of homeostasis in the process of autophagy is essential to preserve stem cell pluripotency. The phenomenon of pluripotency is one of the most important features of cancer stem cells (Figure 5); without it, we would not be able to maintain the cancer stem cells in an undifferentiated state, and it would not be possible to divide the cells continuously [140]. Crucial for the process of autophagy proteins, such as Beclin-1 or Atg4 seem to be essential for maintaining the proper functioning of stem cells. This is confirmed by investigations carried out on cancer stem cells isolated from different types of malignant cancers (liver, pancreas, ovary or breast) [141,142,143,144].

Studies have shown that inhibition of the process of autophagy in cancer stem cells may lead to a decrease in the secretion of IL-6. This process is most likely to occur via the STAT3/JAK2 pathway. It can be concluded that the IL-6-JAK2-STAT3 pathway plays a significant role in the process of conversion of noncancer stem cells into cancer stem cells [145,146]. Many studies also concern FOXO proteins and the role that these transcription factors play in preserving the function of cancer stem cells [147,148,149]. It has been shown that inhibition of FOXO3 protein expression leads to an increase in the potential of cancer stem cells to self-renew. This effect has been observed in breast, prostate or colorectal cancer [150,151,152]. Moreover, many studies indicate the key importance of FOXO proteins to regulate the expression of many proteins directly related to the process of autophagy (for example, Beclin 1, LC3, ULK1 or a number of ATG proteins) [153,154]. One of the recent studies showed that FOXA2 (Forkhead Box A2) protein is overexpressed in cancer stem cells and its activity is directly related to the intensity of autophagy. The Peng team conducted a series of studies to inhibit FOXA2 activity in a cell. In this way, they managed to significantly reduce the ability of cancer stem cells to self-renew [142]. 

## 11. Mitophagy in Cancer Stem Cells

Mitochondria are among the most important cellular organelles. They are responsible for the production of cellular ATP during oxidative phosphorylation and also play a very important role in the production of reactive oxygen species and activation of processes responsible for apoptotic cell death [155,156]. In cancer stem cells, mitochondria play a very important role. The proper functioning of mitochondria determines many properties of cancer stem cells, including their ability to migrate, drug resistance or ability to self-regenerate [157,158,159]. Mitochondria are responsible for cellular respiration. The most of the cancer cells switch to aerobic glycolysis despite the availability of oxygen, which is called the Warburg effect [160,161]. Cancer stem cells are unique in this field. In stem cells exceptional adaptive abilities are observed, depending on the microenvironment in which the cancer stem cells are located, they are able to respirate either by oxidative phosphorylation or perform aerobic glycolysis [157,162,163].

Mitochondria have a very dynamic nature and are subject to the processes of fusion and fission. The fission is regulated mainly by GTPase DRP1 and a number of its associated factors, among others by FIS1. The fusion process is regulated by GTPase MFN1, GTPase MFN2 and OPA1. Both of these processes are carried out in the cell in order to adapt to the current needs of the cell and ensure proper degradation of damaged organelles. Degradation of damaged or redundant mitochondria usually takes place through mitophagy [164,165]. Many studies indicate that the maintenance of both stem cell and cancer stem cell pluripotency depends on the proper functioning of DRP1 regulated mitochondrial fission [166,167,168,169]. The research also confirms that high activity of DRP1 protein may be a bad prognostic factor in the context of cancer treatment [170,171]. 

Mitophagy in the cancer stem cells is usually performed with the PINK1 - PARKIN protein cascade (Figure 6) [172]. The PINK1 protein specifically attaches to the membrane of damaged or dysfunctional mitochondria, and then mediates the phosphorylation of E3 PARKIN ligase. As a result of such changes, the mitochondria intended for degradation are marked with ubiquitin, which in turn allows for their recognition by OPTN, NDP52 and AMBRA1. These proteins further contribute to the process of formation of the autophagosomal membrane around organelles [156]. These processes seem to be crucial for the proper functioning of cancer stem cells [173,174]. The research also shows that mitophagy is regulated in cancer stem cells by increasing the transcription of the NANOG protein, a transcriptional factor that seems to be necessary to preserve the characteristic properties of stem cells, such as the ability to self-renew [175]. The studies also show that cancer stem cells can avoid the activation of apoptosis by specifically activating the mitophagy process. Under conditions of hypoxia (hypoxia characterises most tumours), cancer stem cells switch their cell respiration from oxidative phosphorylation to aerobic glycolysis. As a result, HIF-1α and HIF-2α are activated (these factors are not activated in the cell under normoxia conditions) [176,177]. Under such conditions, cancer stem cells are able to use FUNDC1, BNIP3 and BNIP3L/NIX proteins for energy production and, consequently, significantly reduce the amount of mitochondria in the cell through the mitophagy process. A small amount of mitochondria, in turn, allows to reduce the probability of release of proapoptotic factors from mitochondria and degradation of the whole cell through apoptosis [178,179,180,181].

## 12. The Role of NAD+ and Nicotinamide Phosphoribosyl Transferase (NAMPT) in Induction of Autophagy of Cancer Stem Cells

NAD is a compound that acts as a cofactor necessary for cell metabolism and energy production. It also plays a significant role in the process of DNA repair and maintenance of proper functioning of mitochondria in cancer cells and cancer stem cells (Figure 7). NAMPT, in turn, plays a key role in the synthesis of NAD, after the activity of this transferase is inhibited, a gradual depletion of intracellular NAD+ pool is observed, and thus the ATP synthesis is inhibited [183]. NAMPT is overexpressed in the cancer stem cells, the higher the level of this transferase, the worse is the prognosis in terms of cancer development [184]. Research also shows that NAMPT plays a very important role in regulating the functioning of cancer stem cells, and PARPs and SIRTs also participate in the whole process [185]. It also seems that NAMPT may also constitute one of the factors influencing tumour differentiation (it is responsible for epigenetic reprogramming observed in most cancers), and the activity of this enzyme is regulated by the supply of NAD in the cell [186,187].

## 13. The Importance of Autophagy in Cancer Stem Cell Resistance to Chemotherapy

Despite the constant development of pharmacology and the construction of new, increasingly specific chemotherapeutic schemes, it is still not possible to introduce a therapy that is always effective. This is mainly due to the fact that pharmacological therapy is only able to eliminate cells that are characterised by fast and unlimited proliferation, and that the resting cancer stem cells acquire resistance to the therapy during this time. Such cells are then the source of the recurrence of the cancer, which was previously apparently eliminated.

Cancer stem cells acquire resistance to pharmacological treatment by compiling several factors. First of all, the niche in which these cells reside favours this phenomenon. Another important factor is the high potential to repair DNA damage, inhibition of apoptosis process induction and high expression of the MDR gene [188,189]. The process of autophagy in the cancer stem cells seems to be crucial to maintain the effect of resistance to chemotherapeutics. Many studies show that the combination of autophagic inhibitors and conventional chemotherapeutics allows for a much higher therapeutic index [49,190,191,192,193,194,195]. There is also evidence that the process of autophagy may also have a completely different role in acquiring resistance to chemotherapeutics. For example, in breast cancer stem cells the inhibition of Wnt pathway due to resveratrol activity was observed, which in turn triggered the autophagy process in these cells [196]. Moreover, there is also evidence that inactivation of mTOR may lead to differentiation of cancer stem cells derived from the nervous system [197,198,199]. All this experimental evidence suggests that the process of autophagy may ultimately prove to be one of the key factors responsible for breaking the multidrug resistance of cancer and inhibiting its recurrence.

## 14. The Impact of Autophagy on the Migration of the Cancer Stem Cells

Cancer stem cells have a very high potential for migration and metastasis. This seems to be very strongly associated with EMT (epithelial-to-mesenchymal transition), during which the polarisation of cells and the mechanism by which the cells contact each other are modified [200,201,202,203]. An increasing amount of experimental evidence is now being presented that EMT and autophagy are very closely correlated processes. Cells that have been exposed to EMT are being modified in a specific way. In such cells, the process of autophagy is very intensive [204,205,206].

Some studies have also indicated a completely different effect of autophagic activity in the context of migration. In some tumours, despite the high activity of the autophagic process, low potential of cells for migration has been observed. The modulation of autophagy in such cases allowed to restore the mesenchymal phenotype presentation by the cancer stem cells and, consequently, such cells were capable of migration and metastasis [204,207]. This is most likely related to the phenomenon of paracrine release from the cells, which stimulates EMT induction. Some studies also indicate a completely different effect of autophagic activity in the context of migration. In some tumours, despite the high activity of the autophagic process, low potential of cells for migration has been observed. The modulation of autophagy in such cases allowed to restore the mesenchymal phenotype presentation by the cancer stem cells and, consequently, such cells were capable of migration and metastasis [204,207]. This is most likely related to the phenomenon of paracrine release from the cells, which stimulates EMT induction.

## 15. Conclusions

In many studies, it has been shown that by affecting autophagy, the effectiveness of existing treatment methods can be increased. Nonetheless, there are a few unresolved issues. 

The first one is a comprehensive understanding of interplay between autophagy and cancer resistance [208,209]. There are well known cases where the induction of autophagy inhibited the growth of a tumour. The question is whether inhibition of autophagy is the right solution. Does it depend on the type of tumour? A more precise understanding of the dual role of autophagy would allow for the design of the new systems for cancer treatment [210]. 

Another issue is the development of compounds which would act on autophagy associated with the resistance acquisition. Polytherapies making use of known chemotherapeutics, together with new compounds, are a potential solution [211,212]. Therapy that uses autophagic inhibitors is another unresolved issue, because the side effects of inhibiting autophagy are not well understood. Tissue damage was observed after treatment with autophagic inhibitors, but the appropriate dose of the drug is a great challenge because it depends on the patient [213]. 

Another thing worth considering is which steps of autophagy should be inhibited, i.e., the early stage or the stage of degradation of autolysosome content.

Another problem is the lack of sufficient methods to track autophagy in cancer patients. The only markers that make it possible to monitor autophagy are LC3 and SQSTM1/p62, but they are involved in the process of autophagy [214]. These methods are unable allow to show the activity of the stage of autophagy degradation. Moreover, one cannot forget about post-translation modifications, which have a huge impact on the regulation of autophagy. Therefore, it is very important to find new autophagy markers that do not take part in the autophagy process and include potential posttranslational modifications [74]. 

A further issue is cancer stem cells. CSCs may be closely related to the resistance acquisition by cancer cells. Theoretically, by influencing autophagy and CSCs at the same time, cancer cell sensitisation should be possible and could even lead to overcoming resistance [215,216].

Despite many questions and doubts, it can be concluded that autophagy plays a key role in numerous processes in cancer. It is responsible both for maintaining cell proliferation after chemotherapy and for their elimination. Therefore, it is extremely important to shift these roles of autophagy towards inhibiting cell proliferation. Great hopes are placed on the use of autophagy in inhibiting the transformation of stem cells into cancer stem cells, which may be the key to effective anticancer therapy. That is why it is so important to look for new solutions that will help to modulate this process in such a way that autophagy only plays an antiproliferative role in the context of effective chemotherapy.

## Figures and Tables

**Figure 1 ijms-22-02981-f001:**
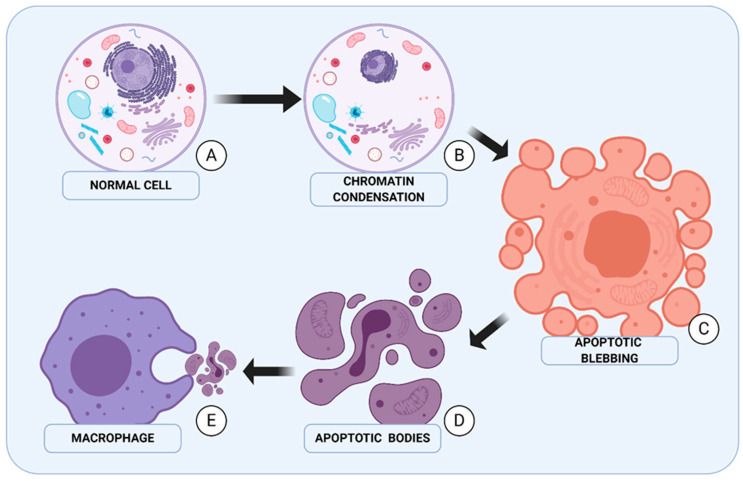
Stages of apoptosis. (**A**) The first morphological symptom of apoptosis is a change at the nucleus level. (**B**) Chromatin undergoes condensation and aggregates under the nuclear membrane, then the nucleus shrinks and undergoes fragmentation. (**C**) Next stage of “dying” is cytoplasm condensation and creation of characteristic bubbles on the cell surface. (**D**) Apoptotic bodies are constructed of cell fragments and consist of organelles, cytoplasm and chromatin. (**E**) The final stage of apoptosis is apoptotic bodies phagocytosis.

**Figure 2 ijms-22-02981-f002:**
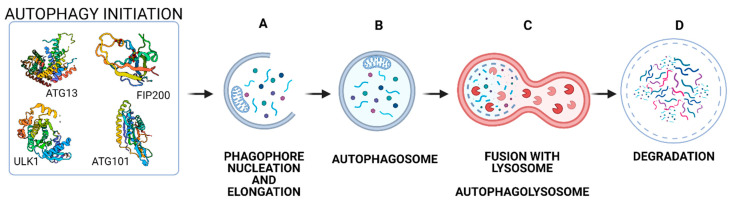
Various signals activate autophagy. The target of these signals is ULK 1 complex, which includes ULK1, FIP200, ATG13 and ATG101 protein. During autophagy fragment of cytoplasm is surrounded by C-shaped double-membrane (**A**), while autophagosome is being created as the result of connection of two membrane ends (**B**). Afterwards autophagosome fuses with lysosome. The resulting vesicle is known as an autophagolysosome (**C**). In this vesicle the degradation of macromolecular substrates for into their basic constituents occurs (**D**).

**Figure 3 ijms-22-02981-f003:**
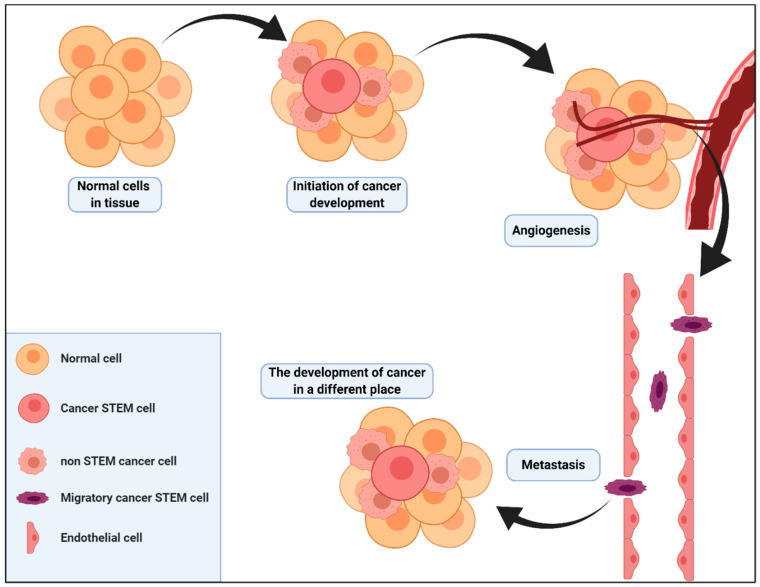
The formation of neoplastic stem cells and the mechanism of cancer formation and its metastases. The development of a tumour is usually sequential. After the appearance of the first neoplastic cells, i.e., both stem cells and nonstem cells, the tumour develops gradually. In the next stage, the process of angiogenesis begins, and as a result of the formation of blood vessels, the invasive cells can move to other regions of the body.

**Figure 4 ijms-22-02981-f004:**
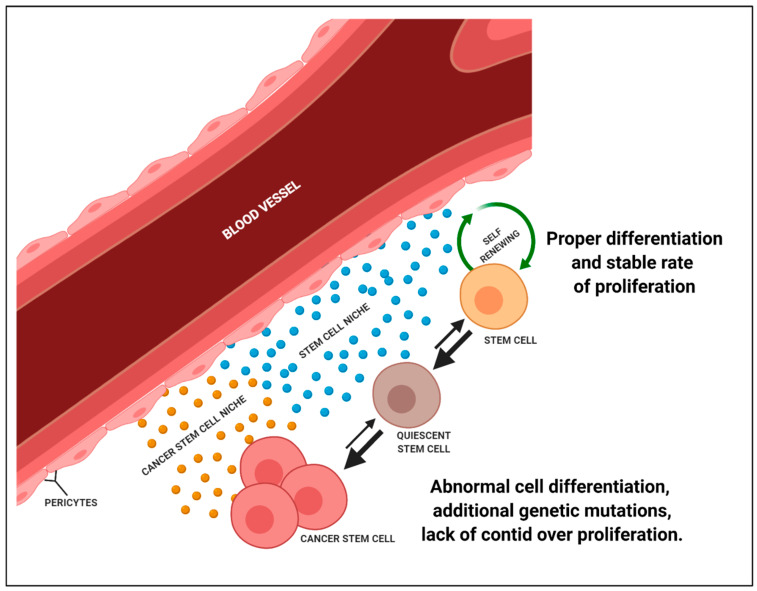
Comparison of functionality of normal stem cell niche and cancer stem cells niche. A niche of stem cells under physiological circumstances provides the cells with conditions in which both cell differentiation and cell proliferation are inhibited. Only during the regeneration of the tissue in which the stem cells are located, a transient signal for proliferation is activated. When appropriate mutations of genetic material develop in the stem cell, they start to proliferate in an uncontrolled way. Another hypothesis assumes that as a result of such a mutation, the niche of the stem cells provides uninterrupted signals that initiate the proliferation and growth of the stem cells. As a result, a large pool of progenitor cells is created, which have a genetic mutation.

**Figure 5 ijms-22-02981-f005:**
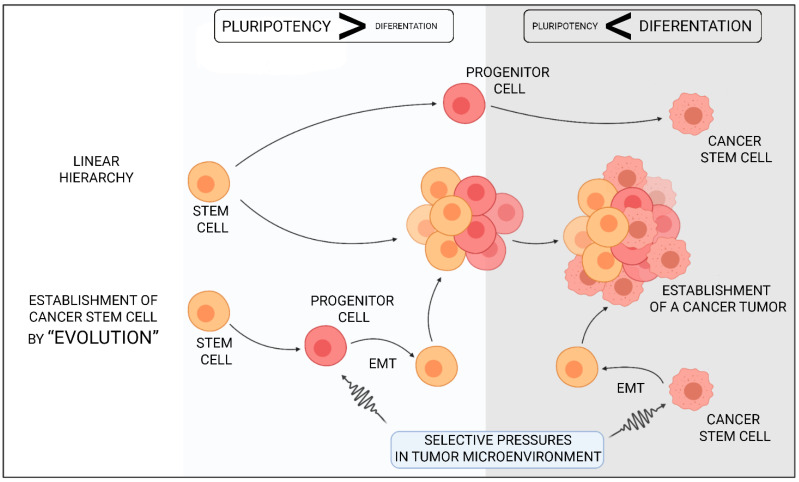
Two basic theories on the formation of cancer stem cells. In the theory assuming a hierarchical model of CSC formation, it is genetic and epigenetic transformation that leads to the formation of progenitor cells and cancer stem cells. The cells processed in this way undergo further transformations, which ultimately lead to the development of a tumour. Another theory, assuming the “evolution” of cancer stem cells, assumes that the whole process begins by induction of EMT, either as an element of the development of the disease or as a result of the activity of selected signals coming from the microenvironment of the tumour. Epithelial-to-Mesenchymal transition (EMT) is a natural process that triggers the development of the tumour, it determines changes in cell polarity and cell-cell contact.

**Figure 6 ijms-22-02981-f006:**
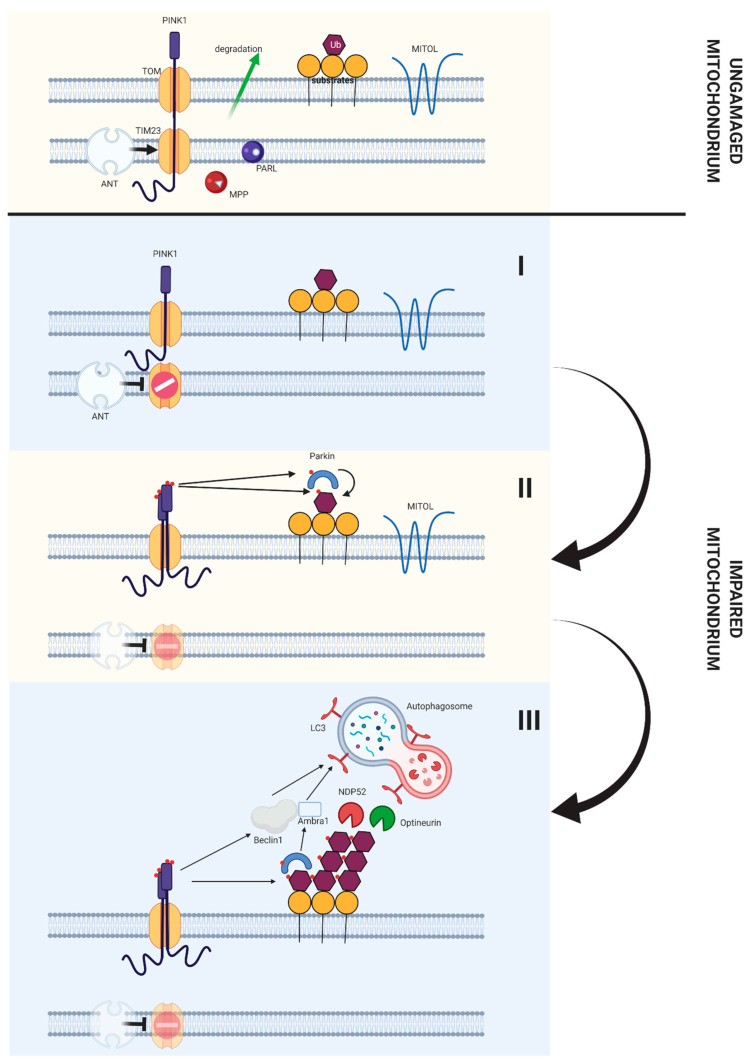
Role of the PINK1/Parkin pathway in induction of autophagy both in stem cells and differentiated cells. In a properly functioning mitochondria PINK1 is transported through the TOM/TIM23 translocase complex to the internal mitochondrial membrane. It is then disintegrated (due to the activity of two proteases, PARL and MPP), transported to cytosol and degraded. When mitochondria do not functioning properly (which manifests itself as a dysfunction of mitochondrial membrane potential), ANT inhibits the transport of PINK1 through the TIM23 translocase. As a result of this phenomenon PINK1 accumulates on the external mitochondrial membrane. Accumulation of PINK1 on the external mitochondrial membrane results in formation of complex consisting of PINK1 and TOM translocase, which results in autophosphorylation of PINK1 and its activation. The activated PINK1 phosphorises the ubiquitinated substrates located on the external mitochondrial membrane, which makes it possible to connect Parkin and its phosphorylation. Phosphorylated Parkin, in turn, contributes to the ubiquitination of subsequent proteins on the external mitochondrial membrane. The ubiquitinated proteins, in the other hand, are able to phosphorylate subsequent PINK1 molecules, thus creating a feedback loop, so that a malfunctioning mitochondrion is covered with phosphoubiquitin. Then mitophagic adapters Optineurin and NDP52 bind phosphoubiquitin chains. These proteins further bind the LC3 protein, which allows the initiation of mitophagy. At the same time PINK1 interacts with Beclin-1, Parkin and Ambra proteins, which contributes to the formation of autofagosome around the damaged mitochondria [182].

**Figure 7 ijms-22-02981-f007:**
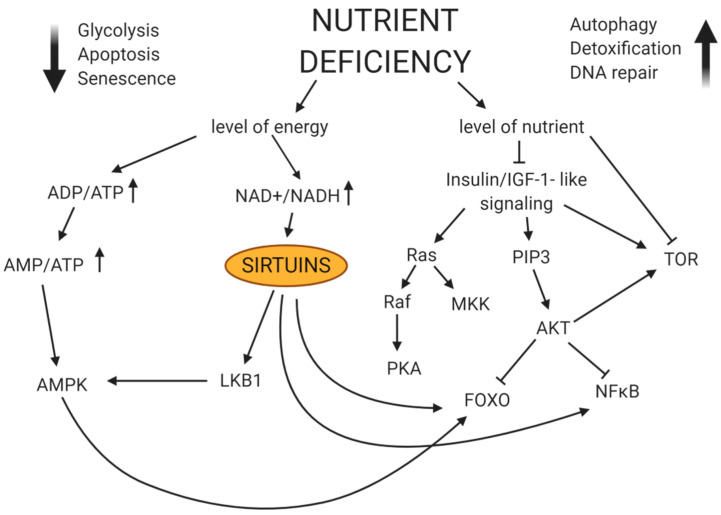
The influence of nutrient availability on the induction of autophagy. The decrease in availability of nutrients contributes to a decrease in mitochondrial activity, which in turn increases the ratio of AMP to ATP and contributes to an increase in intracellular NAD+ levels. The consequence of increasing the AMP to ATP ratio is the activation of AMPK. Moreover, AMPK is activated by deacetylation, which generates a feedback loop. Increased NAD+ level, in turn, increases sirtrulines activity, which causes deacetylation of LKB1 and AMPK and regulation of their functions. A downwards pointing arrow shows a decrease in the intensity of the process. An upward arrow indicates an increase in the intensity of the process described.

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
