# Peer review of "Two Faces of Autophagy in the Struggle against Cancer"

_ijms, 2021, doi:10.3390/ijms22062981_

Round 1
Reviewer 1 Report
The review article entitled" Between autophagy and apoptosis in the fight against cancer" describes the relationship between autophagy and apoptosis and their role in cancer progression. Abstract covered main contents properly. The introduction is well organized with sufficient literature review. The findings summarized here are very important to the scientific community and use in new drug development. The authors clearly discussed apoptosis, autophagy, and their mechanisms and discussed their roles in cancer. However, there are some minor concerns to answer before considering further.
- The title is not fitting correctly here, suggesting that authors rewrite the title with more appropriate meaning.
- The legibility of figure 2 is poor, suggesting to increase the legibility of figures 2
- Figure 3 can be modified furthermore
- all the figures quality can be improved
Reviewer 2 Report
The paper is of interest and well written.
The paragraph about autophagy inducers could be improved, maybe rapamycin and metformin should be further discussed.
Figure 3 is probably irrelevant, I would delete it.
